# Seventy Years of Antipsychotic Development: A Critical Review

**DOI:** 10.3390/biomedicines11010130

**Published:** 2023-01-04

**Authors:** Mujeeb U. Shad

**Affiliations:** 1UNLV School of Medicine, University of Nevada, Las Vegas, NV 89154, USA; mujeebushad@gmail.com; 2College of Osteopathic Medicine, Touro University Nevada, Las Vegas, NV 89014, USA; 3Psychiatry Residency Program, Graduate Medical Education, The Valley Health System, Las Vegas, NV 89118, USA

**Keywords:** seventy, years, antipsychotic, development, critical, review

## Abstract

Since the mid-1950s discovery of the first effective antipsychotic medications (APM), we have only been able to improve the tolerability but not the overall efficacy of currently available APMs, as reflected by effectiveness trials in Europe and the United States. This inability to develop more effective APMs is attributable to multiple factors, including failure to create and use assessment tools to assess core symptom domains in schizophrenia, move beyond the dopaminergic hypothesis and to develop “me too” drugs, imposing ill-defined research domain criteria, and lacking federal funding for clinical trials. The classification of APMs is also confusing, including second-generation, partial agonists, and multimodal APMs in the same class of APMs, despite significant differences in their mechanisms of action. Other factors stagnating drug development include inadequate sample sizes to address heterogeneity, lack of statistical measures correlating with clinical significance, using the atheoretical basis of psychiatric diagnoses, failure to control placebo response, and high cost of newer and perhaps more tolerable APMs. Furthermore, there has been a failure to develop early predictors of antipsychotic response and various tools to optimize an APM response. Finally, some mental health providers are also responsible for the suboptimal use of APMs, by using excessive maintenance doses, often with irrational polypharmacy, further compromising effectiveness and medication adherence. However, some bright spots in antipsychotic development include improved tolerability of APMs and long-acting injectables to address the high prevalence of medication nonadherence. This review critically reviews 70 years of antipsychotic development, the reasons behind the failure to develop more effective APMs, and suggestions for future direction.

## 1. Introduction

While pharmacotherapy of depression appears to have moved from the monoamine hypothesis to more fertile grounds of glutamatergic and GABAergic mechanisms, it has been challenging to shift the antipsychotic paradigm beyond the dopaminergic hypothesis of schizophrenia. Efforts to develop non-dopaminergic antipsychotic medications (APMs) have produced negative results; thus, there has been no effective APM without dopamine involvement during 70 years of antipsychotic drug development. The only exception has been the approval of pimavanserin, a selective 5HT2A receptor blocker, to treat Parkinson’s psychosis but not schizophrenic psychosis.

Although Federal Drug Agency (FDA) has approved multiple APMs in the last 70 years, most have been with modest variations in molecular targets to qualify for “me too” drugs. The lack of significant differences in mechanisms of action explains why current APMs are primarily effective for positive symptoms without any measurable improvements in negative and cognitive symptoms. The antipsychotic response has been defined as only a 20% reduction in total scores on the most commonly used scale to assess schizophrenia symptoms, the Positive and Negative Syndrome Scale (PANSS) [1]. This reduction in response criterion accommodated the limited efficacy of current APMs for negative and general schizophrenia symptoms, including some cognitive symptoms, such as orientation, attention, insight, and judgment. However, a 20% reduction in total PANSS scores may be mediated by a completely different set of schizophrenia symptoms from one to the other APM, making it difficult to interpret and compare antipsychotic efficacy with precision.

Major psychiatric disorders are diagnosed at a syndromic level, with various symptom domains having potentially different neurobiological underpinnings. Still, most psychiatric studies have used atheoretical diagnoses based on the Diagnostic Statistical Manual [2,3] or the International Statistical Classification of Diseases and Related Health Problems (https://icd.who.int, accessed on 6 December 2022), which could explain the failure to develop comprehensive treatments. In addition, inadequate sample sizes have failed to account for heterogeneity across schizophrenia patients, not allowing for sub-group analyses to generate testable hypotheses and guide future research. These diagnostic limitations were addressed by restricting federal funding for prespecified Research Domain Criteria (RDoC) [4], which were vague, poorly defined, and not scientifically rigorous. Unfortunately, this measure was also counterproductive in promoting neurobiological research that could help advance psychiatric treatments.

Furthermore, researchers have failed to develop early predictors or intermediate phenotypes for antipsychotic response or tolerability, which could guide treatment options for individual patients. In addition, there is almost no research on finding the maintenance dose and length of antipsychotic therapy in schizophrenia patients recovering from acute psychosis. Similarly, there is a lack of laboratory measures to optimize antipsychotic response or tolerability, except for hyperprolactinemia (indicating excessive D2 receptor occupancy), therapeutic drug monitoring (TDM) for clozapine, and pharmacogenetic testing for refractory patients. Even though these measures can enhance antipsychotic efficacy and tolerability, particularly in treatment-refractory schizophrenia, they are not frequently employed, resulting in suboptimal use of APMs [5]. In addition, some providers use high antipsychotic doses, especially in severely and persistently ill patients in state hospitals and prison systems, compromising tolerability and medication adherence and resulting in repeated hospitalizations [5]. Furthermore, the antipsychotic doses required to manage acute psychosis and behavioral dyscontrol are not reduced even after the patient stabilizes. Several clinicians do not employ therapeutic drug monitoring for clozapine before labeling patients as clozapine non-responders [6].

Since NIMH does not fund clinical trials, developing new APMs is primarily owned by the pharmaceutical industry. Although this process involves billions of dollars and may not be fundable by the NIMH, post-marketing and repurposing trials for already approved agents can be affordable, providing a less biased approach than industry-sponsored trials. Another limitation is that the results from the preclinical trials for drug approval cannot be applied to the general population as they are completed in a near-ideal patient population without comorbidities. Thus, it takes years to gather post-marketing effectiveness data applicable to the general patient population. Furthermore, prescriptions of newly approved APMs are too expensive to be covered by Medicaid to be used in the severely and persistently ill, homeless, and forensic population that needs these medications the most.

Although a detailed discussion of the statistical limitations of clinical trials is beyond the scope of this review, the statistics using *p* < 0.05 has failed to translate into clinically relevant results [7]. However, this trend is changing to more clinically applicable measures, such as confidence interval, effect size, number needed to treat (NNT) [8], and number needed to harm (NNH) [9]. In addition, an increasing level of placebo response may have also failed several promising psychotropics, including APMs [10]. In this context, a higher placebo response can lower the chances of finding a significant difference (i.e., signal) between placebo and active treatment. Several strategies have been proposed to reduce placebo response, but none have been foolproof [11].

However, despite all these obstacles in developing more effective APMs, currently available treatments have made some progress in tolerability, if not efficacy. Thus, relatively higher dose thresholds for adverse effects are observed with second-generation APMs, partial agonists, and especially the recently approved multimodal antipsychotic lumateperone. It is also worth mentioning that long-acting injectables (LAIs) have significantly impacted long-term clinical and functional outcomes in schizophrenia patients worldwide. This review provides a synopsis of different mechanism-based classes of antipsychotic medications approved to manage schizophrenia and other psychotic disorders over the last 70 years.

While pharmacotherapy of depression appears to have moved from the monoamine hypothesis to more fertile grounds of glutamatergic and GABAergic mechanisms, it has been challenging to shift the antipsychotic paradigm beyond the dopaminergic hypothesis of schizophrenia. Efforts to develop non-dopaminergic antipsychotic medications (APMs) have been disappointing, with none of the currently available APMs moving beyond the dopamine system during 70 years of antipsychotic drug development. The only exception has been the approval of pimavanserin, a selective 5HT2A receptor blocker, to treat Parkinson’s psychosis but not schizophrenic psychosis.

Although Federal Drug Agency (FDA) has approved multiple APMs over the last seven decades, most have been with modest variations in molecular targets to qualify for “me too” drugs. The lack of significant differences in mechanisms of action explains why current APMs are primarily effective for positive symptoms without any clinically meaningful improvements in negative and cognitive symptoms. Therefore, the antipsychotic response is defined as only a 20% reduction in total scores on the Positive and Negative Syndrome Scale (PANSS) [1] instead of a 50% reduction required for an antidepressant response. This reduction in response criterion accommodated the limited efficacy of current APMs for negative and some of the cognitive symptoms assessed with PANSS, such as orientation, attention, insight, and judgment. A 20% reduction in PANSS scores suggests an antipsychotic response may be defined by a completely different set of symptoms from one patient to the other, making it difficult to interpret and compare antipsychotic efficacy with precision.

Major psychiatric disorders are diagnosed at a syndromic level, with various symptom domains having potentially different neurobiological underpinnings. However, most diagnoses in psychiatric research are based on atheoretical diagnostic tools such as the Diagnostic Statistical Manual [2,3] or the International Statistical Classification of Diseases and Related Health Problems (https://icd.who.int, accessed on 6 December 2022). The lack of neurobiologically-oriented diagnoses could explain the failure to develop more effective and comprehensive treatments. In addition, relatively small sample sizes have failed to account for heterogeneity across schizophrenia patients, not allowing for sub-group analyses to generate testable hypotheses and guide future research. The relative failure of DSM-based research was addressed by restricting federal funding to prespecified Research Domain Criteria (RDoC) [4], which were also vague, poorly defined, and not scientifically rigorous. Unfortunately, this measure was also counterproductive in promoting neurobiological research that could help advance psychiatric treatments.

Thus, second-generation APMs, including partial agonists, may have higher dosing thresholds to cause adverse effects, such as extrapyramidal symptoms (EPS) and hyperprolactinemia. It is also worth mentioning that long-acting injectables (LAIs) have significantly impacted long-term clinical and functional outcomes in schizophrenia patients worldwide. This review provides a synopsis of different mechanism-based classes of antipsychotic medications approved to manage schizophrenia and other psychotic disorders over the last 70 years. However, summarizing the role of APMs in managing bipolar disorder is beyond the scope of this review.

## 2. Older Antipsychotic Medications

Despite limited efficacy and frequent adverse effects, older or conventional APMs provided the first effective treatment for schizophrenia patients, who did not have any chance to live outside asylums and the prison systems. These APMs are further subclassified into high potency and low potency. The low-potency conventional APMs are generally less tolerable than the high-potency conventional APMs, due to multiple sites of action representing a “shotgun” approach [12]. These actions include muscarinic, histaminic, and alpha-1 adrenergic receptor blockade. Antimuscarinic effects include dry mouth, blurred vision, urinary retention, constipation, tachycardia, loss of sweating, confusion, and worsening of closed-angle glaucoma and cognitive function. Antihistaminic effects translate into sedation and short-term weight gain, and the alpha-1 adrenergic blockade mediates postural hypotension, dizziness, and sedation.

In contrast, high-potency APMs are relatively more selective for blocking dopamine-2 (D2) receptors, and the main adverse effects associated with them are due to D2 blockade, especially at high doses. These adverse effects include extrapyramidal symptoms (EPS), hyperprolactinemia (Figure 1 and Figure 2), and treatment-induced, or worsening of, pre-existing negative, affective or cognitive symptoms.

However, the most clinically-serious adverse effect of low-potency APMs is the effects on QTc prolongation, which can result in sudden cardiac death [13]. Although the high-potency APMs also prolong QTc interval, the most clinically serious effects include tardive dyskinesia [14] and neuroleptic malignant syndrome [15]. These adverse effects are generally reported with long-term use of high-dose antipsychotic pharmacotherapy [15]. However, some of the adverse effects of low-potency conventional APMs may be beneficial in few patients, including sedation and weight gain with the histamine-1 receptor blockade, reduction in pre-existing hypertension with the alpha-1 receptor blockade, and protection from EPS with the built-in anticholinergic effects. These adverse effects are also witnessed with newer APMs, such as olanzapine and clozapine [16]. Of note, none of the APMs is indicted for managing dementia-related psychosis due to increased risk for mortality in older adults [17].

There is one APM, loxapine, that is not only effective in schizophrenia but also in major depressive disorder because loxapine is demethylated to a tetracyclic metabolite, amoxapine, which has antidepressant properties [18]. Another worth-mentioning APM, molindone, has a moderate affinity for D2 receptors, with potential benefits in patients not responding to high-or low-affinity APMs [19]. It is also interesting that one of the older high-potency APM, pimozide is the only FDA-approved treatment for Tourette’s syndrome [20].

One of the most clinically significant advances in antipsychotic treatments has been the development of long-acting injectables (LAI) with high-potency APMs, haloperidol, and fluphenazine, followed by newer APMs, risperidone, paliperidone, and aripiprazole. Since medication nonadherence is commonly observed in schizophrenia subjects, the LAIs have improved the maintenance of antipsychotic response and prevention of psychotic relapse and rehospitalizations [21]. In addition, these LAIs are now also available with some of the newer generation APMs as discussed below.

In summary, serotonin and dopamine antagonism (SDA) with SGAPMs may have provided some cushion from dose-related adverse effects but without any significant benefits in efficacy. The only exception among SGAPMs is clozapine, which remains the gold standard for managing treatment-refractory schizophrenia (TRS). However, despite its unique efficacy, to our knowledge, clozapine has not been compared with other SGAMs or studied in post-marketing effectiveness or efficacy trials because it is only approved for patients with TRS.

## 3. Second-Generation Antipsychotic Medications (SGAPMs)

This class includes every APM developed after the conventional APMs despite significant differences in the mechanisms of action (Table 1), which has created confusion and misperceptions.

In addition, assigning clozapine to the newer APMs is not accurate, as clozapine was approved for clinical use in 1971 [31], even before the discovery of some of the older or first-generation APMs. However, it was not until 1989 that the FDA approved clozapine [31], perhaps due to the risk of bone marrow toxicity resulting in an earlier discontinuation in Europe. Similarly, the older versus new classification of APMs is not such a black-and-white concept. One of the older APM, loxapine, has been found to have more potent D2 than the 5HT2A receptor blockade, especially at lower doses (Table 1) [32], a profile that distinguishes older from newer APMs [33]. Risperidone, one of the earliest atypical after approval of clozapine, offers a similar challenge as risperidone loses its atypicality at higher doses and becomes a more potent D2 than 5HT2A receptor blocker [33]. At higher doses, risperidone has been shown to have an even higher level of prolactin (a surrogate for D2 receptor occupancy) than the prototypical high-potency APM, haloperidol [34].

Even within the second-generation APMs, there are significant differences despite their classification in the same group. Starting with clozapine is still one of the unique APM and the only one with proven efficacy in treatment-refractory schizophrenia (TRS) and anti-suicidal effects [35]. If there was no bone marrow toxicity, clozapine could be initiated early as the first line of treatment, improving clinical outcomes and prognosis at the onset of first-break psychosis. However, clozapine is not an easy APM to use, and several providers in the United States remain hesitant to initiate clozapine treatment even in TRS, explaining the underutilization of clozapine treatment in the United States among most developed countries in the world [36]. Clozapine’s toxicity to the neutrophils, which may result in agranulocytosis, explains why the regulatory authorities require registration with the Risk Evaluation and Mitigation Strategy (REMS) to monitor WBC count before a pharmacy can prescribe clozapine for the patients. Unfortunately, this blood test requirement also appears to make some patients hesitant to use clozapine treatment due to fear of needles. However, some of the needle-associated fear is overblown, and if patients are adequately educated about the risks and benefits of clozapine, a significantly large number may agree to use clozapine. At the same time, some European countries, such as the UK, have been adopting increasingly less stringent interval requirements for repeating blood tests required for clozapine treatment (http://www.medicines.org.uk/emc/medicine/1277, accessed on 8 December 2022). This is because agranulocytosis decreases from 0.70/1000 patient-years in the second 6 months of treatment and, after the first year, 0.39/1000 patient-years [37]. Nevertheless, few cases of late-onset clozapine-induced agranulocytosis have been reported [38], requiring continued WBC monitoring. There are other clinically serious effects of clozapine, including myocarditis [39], megacolon, and lowering of the seizure threshold [40]. Another adverse effect that requires special mention is the frequently reported clozapine-induced drooling (sialorrhea) [41]. If not properly treated, it can significantly decline oral hygiene and negatively impact the quality of life. In contrast to all these adverse effects, EPS and hyperprolactinemia are relatively rare with clozapine, probably due to its low water dissociation constant and transient and loose binding affinity for D2 receptors (Table 1) [42]. Therefore, even doses around 900 mg/day of clozapine have not been reported to increase EPS and prolactin, adverse effects that are primarily mediated by D2 receptor blockade. This characteristic is unique to clozapine and another newer APM, quetiapine [43].

Although clozapine is the only effective APM in TRS, not all treatment-refractory patients are clozapine responders [44]. However, there may be many TRS patients who may not be genuine nonresponders to clozapine and may be potentially converted into responders or partial responders by utilizing laboratory tools, such as therapeutic drug monitoring (TDM), to optimize clozapine response [6]. Clozapine has multiple metabolic pathways provided by cytochrome P-450 (CYP) enzymes, CYP1A2, CYP2D6, and CYP3A4 [45]. Genetic polymorphism in these enzymes may alter the efficacy and/or tolerability of any APM metabolized by these enzymes, including clozapine [46]. The ratio between clozapine and its primary metabolite, norclozapine (Clz/Nclz), is one of the best clinical examples of using TDM to predict the activity of CYP1A2 [6]. Since clozapine is biologically different from its primary metabolite, Clz/Nclz ratio may provide some insights into clozapine’s efficacy, safety, and tolerability [47]. A higher Clz/Nclz ratio suggests a low activity of CYP1A2, perhaps due to drug interactions, and a low ratio may denote the increased activity of CYP1A2 resulting in faster conversion of clozapine to norclozapine, which can happen with smoking [48]. Case studies have reported the lower threshold for clinically effective clozapine levels is 350 ng/mL [49,50,51]. The upper limit ranges between 450 ng/mL and 600 ng/mL, but levels at or above 1000 ng/mL can be toxic and should be avoided [52]. If these measures fail to increase clozapine response, some augmentation strategies can be effective, such as lamotrigine and aripiprazole at low doses [53].

All other SGAPMs do not appear to have the same efficacy profile as clozapine, despite some sharing similar but not identical adverse effect profiles. Olanzapine has some structural resemblance to clozapine, with a similar magnitude and frequency of weight gain but not its efficacy (Figure 3) [54,55].

Nevertheless, olanzapine continued to be taken by the study subjects the longest, despite no differences in efficacy in the large effectiveness trial with APMs, Clinical Antipsychotic Trials of Intervention Effectiveness (CATIE) [56]. This study did not find any meaningful differences in efficacy between older (i.e., perphenazine) and newer APMs [57]. Although the FDA has approved a long-acting formulation of olanzapine, it is rarely used in clinical practice due to post-injection delirium sedation syndrome [58]. Despite multiple efforts, no significant breakthroughs have been achieved in reducing weight gain and metabolic syndrome with olanzapine and clozapine. However, the concomitant use of anti-diabetic medication, metformin, has helped reduce weight gain in some patients [59]. The recent approval of a fixed-dose combination of samidorphan, a mu-opioid receptor antagonist, and the second-generation antipsychotic drug, olanzapine, by the FDA has provided another option for managing olanzapine-induced weight gain [60,61]. This study reported 18% of patients gaining ≥ 10% weight in the olanzapine-samidorphan group as opposed to 30% in the olanzapine monotherapy group (Figure 3) [60].

The following SGAPM, quetiapine, has one of the lowest and most transient binding affinities for D2 receptors (Table 1) [43], to the extent that the preclinical trials showed even lower prolactin levels than the placebo [62]. Although some of this profile may be similar to clozapine, which it structurally resembles, quetiapine has not demonstrated that level of efficacy at the daily recommended dose of 300 mg [63]. Post-marketing data show that quetiapine doses between 600 to 900 mg a day may have better efficacy [63]. Nevertheless, quetiapine has offered a relatively benign option for managing secondary psychosis, especially in Parkinson’s disease [64] and dementias [65], where only a transient dopamine blockade may be desirable. Interestingly, a recently approved medication, pimavanserin for Parkinson’s disease psychosis, is the first treatment for secondary psychosis without direct dopamine involvement [66]. Interestingly, quetiapine has been one of the most frequently prescribed SGAPMs but at lower doses that are only effective for sedative effects and not psychosis [67].

The next APM, ziprasidone, is the first to report near weight-neutral effects and perhaps some reduction in low-density lipids and triglycerides as opposed to some of the other SGAPMs [68]. However, some initial concerns with the impact of QTc prolongation may have limited ziprasidone’s use in clinical practice (Figure 4) [69].

In addition to having less risk for metabolic syndrome, ziprasidone has also been one of the first lines of treatment for agitation and aggression, primarily due to a rapid onset of antipsychotic effects [70] without sedation [71], making it possible to interview an awake patient in the emergency department. In addition, ziprasidone may have some stimulating effects that may explain a few reports of manic conversions, perhaps due to the moderate blockade of norepinephrine and serotonin transporters [72]. These diverse effects of ziprasidone are probably due to its chimeric structure, which means that ziprasidone has two different moieties within the same molecule, resulting in 5HT2A inverse agonism, D2 receptor antagonism, 5HT1A receptor agonism, and a moderate serotonin and norepinephrine pump blockade [73]. This receptor action profile of ziprasidone may be responsible for its efficacy for depressive and negative symptoms in schizophrenia and schizoaffective disorders with fewer adverse effects, particularly EPS, hyperprolactinemia, and metabolic effects in comparison with conventional antipsychotics (Table 1; Figure 1, Figure 2 and Figure 3) [74,75,76]. Ziprasidone is primarily metabolized in the liver with a half-life of about 7 h at recommended doses. There is a low risk of pharmacokinetic drug interactions with ziprasione, as less than 1/3rd of ziprasidone’s metabolism is mediated by CYP enzymes [77,78]. However, ziprasidone must be given with food for effective absorption [79].

Paliperidone is a metabolite of risperidone (i.e., 9-OH risperidone) with a stronger binding affinity for D2 receptors than risperidone (Figure 1 and Figure 2) [30]. However, the higher affinity of paliperidone for D2 receptors does not result in any significant increase in prolactin elevation or EPS with paliperidone as compared to risperidone (Table 1). However, paliperidone has a longer half-life than risperidone and thus can be given once a day orally [30]. Since paliperidone is already a metabolite [80], it may be a safer option than other APMs in patients with liver dysfunction or polypharmacy due to the lower risk for drug interactions [81]. However, the most effective use of paliperidone has been its long-acting injectable (LAI), which provides the most flexibility regarding the duration of effects and LAI frequency. Paliperidone is now available as a monthly (Invega Sustenna), quarterly (Invega Trinza), and biannually LAI (Invega Hayfera), which significantly minimizes the number of patient visits to the clinic and can be cost-effective [82].

Iloperidone, the next SGAM, is one of the most potent blockers of noradrenergic α_1_ receptors (Table 1), which may expose schizophrenia subjects to significant postural hypotension and dizziness and requires a slow titration to avoid these adverse effects [83]. However, a potent noradrenergic α_1_ receptor antagonism may explain one of the lowest risks for akathisia (Figure 2), which, unlike other EPS, may be mediated by an increase in noradrenergic neurotransmission [84]. In addition, iloperidone may become a drug of choice if the posttraumatic disorder is associated with psychosis, as α_1_ receptor antagonism is the putative mechanism behind the efficacy of prazosin in improving sleep efficiency [85]. Similarly, iloperidone can be beneficial in managing comorbid hypertension in psychotic or posttraumatic patients [86]. Thus, overall, iloperidone may have one of the lowest risks of causing akathisia [84]. As can be seen in Figure 3, weight gain with iloperidone is also in the moderate range comparable with risperidone [84]. However, it does have some risk for QTc prolongation [87] (Figure 4).

Asenapine is the only SGAM approved as sublingual preparation and has been recently available as a transdermal treatment in the United States. The sublingual administration is reported to result in a more rapid onset of antipsychotic effects than orally administered APMs and, similar to the transdermal application, delivers a higher proportion of parent drug unaltered by first pass effect [88]. Unlike some of the other SGAPMs, asenapine does not require dose titration to avoid adverse effects, and starting dose can also be the effective dose and can be given once a day [89]. Weight gain is also less than olanzapine and risperidone (Figure 3) [89]. There has also been a suggestion that since asenapine antagonizes 5HT7 and adrenergic α_2_ receptors (Table 1), it may augment antidepressant response [90] and may have cognitive benefits [91].

Lurasidone followed the approval of asenapine as the next SGAPM. Food can affect the absorption of lurasidone but not as much as with ziprasidone [79]. Nevertheless, lurasidone exposure is increased twofold and maximum concentration is up to threefold when administered with food. Like asenapine, lurasidone also has a strong affinity for serotonin 5HT7 receptors (Table 1), which may have implications for cognition [92]. In addition, lurasidone is also a partial agonist at 5-HT1A; this may add further to the procognitive as well as potential antidepressant effects [93]. However, unlike iloperidone, lurasidone has minimal affinity for alpha-1 noradrenergic receptors and a low propensity for orthostatic hypotension (Table 1) [93]. More importantly, lurasidone has minimal affinity for 5HT2C receptors and virtually no affinity for histamine H1, predicting a low liability for weight gain (Figure 3) [94]. No affinity for histamine H1 predicts a lower risk for sedation compared with other APMs with potent histamine blockade (Table 1). Unlike olanzapine and clozapine, lurasidone’s lack of affinity for cholinergic M1 receptors would predict a low propensity for causing anticholinergic side effects (Table 1) [93]. Lurasidone’s dopamine D2 receptor occupancy at 40 mg/day ranges between 63% to 67%; assuming this is within the range of effective D2 receptor occupancy, an antipsychotic response could be potentially achieved with an oral dose of 40 mg/day [95], particularly in elderly subjects. The primary metabolic pathway for lurasidone is CYP3A4 [96], which has clinical implications regarding the use of lurasidone in the presence of an extensive list of inducers and inhibitors of CYP3A4.

## 4. Partial Agonists

Partial agonists of dopamine-2 (D2) receptors are also classified as the SGAPMs that are full antagonists of D2 receptors, which raises some questions. A partial agonist for D2 receptors is an entirely novel concept for which Arvid Carlsson received a Nobel prize in 2000 [97]. Often partial agonism is misperceived as an antipsychotic that only partially activates D2 receptors, which is far from the truth. A partial agonist’s action depends on the baseline neurotransmitter activity at its receptors, so a partial agonist will reduce an increased baseline receptor activity and increase a reduced baseline receptor activity, suggesting that a partial agonist is actually a system stabilizer [97,98]. Thus, a partial agonist at D2 receptors increases low dopamine activity in the mesocortical tract, which may potentially benefit effective, negative, and cognitive symptoms while reducing dopamine hyperactivity in the mesolimbic pathway required for antipsychotic effects [98,99]. In contrast, a full D2 receptor antagonist decreases dopamine activity globally to manage psychosis, including the hypoactive dopaminergic pathways, such as the mesocortical tract, thus increasing the risk for iatrogenically-induced affective, affective, or cognitive symptoms, especially with the high-dose antipsychotic pharmacotherapy [100]. Therefore, partial agonism brings a novel concept that is theoretically consistent with the basic pathophysiological model of schizophrenia with differential dopaminergic activity across dopaminergic pathways. Although this concept may not be clinically superior in managing positive symptoms, the adverse effect profile, particularly concerning EPS and hyperprolactinemia, may be relatively better than the previous classes of APMs, including the SGAPMs [101]. Thus, partial agonists appear to have a more favorable tolerability profile for adverse effects due to D2-receptor blockade, such as EPS and hyperprolactinemia [101]. One of the most significant differences between D2 receptor partial agonism and D2 receptor antagonism is the relative lack of dose-related adverse effects except for akathisia. However, like other APMs, partial agonists have also been associated with weight gain [102]. There are three partial agonist APMs approved by the FDA, aripiprazole, cariprazine, and brexpiprazole.

Aripiprazole was approved in 2002, even before some of the SGAPMs were approved. Although aripiprazole is a partial agonist, it has one of the most potent affinities for D2 receptors [98,99]. But since it is a system stabilizer and not a full antagonist at D2 receptors, the adverse effects and perhaps the antipsychotic efficacy may not always be dose-related. Lack of prolactin elevation and absence of EPS except for akathisia, irrespective of the dose, supports the partial agonism concept with aripiprazole (Figure 1 and Figure 2) [103,104]. Another benefit of aripiprazole and other partial agonists is a relatively lower incidence of weight gain and metabolic syndrome (Figure 3) [105]. Since aripiprazole is a partial agonist for D2 and the 5HT1A receptors, aripiprazole stabilizes two critical neurotransmitter systems in the brain, dopamine, and serotonin, which explains aripiprazole’s efficacy across several psychiatric disorders (Table 1) [106]. A partial agonist, such as aripiprazole, can also be helpful at a relatively low dose to reduce long-term adverse effects in patients receiving LAI, such as EPS and hyperprolactinemia [107]. Aripiprazole is the only partial agonist available as a long-acting injectable (LAI; Abilify Maintena and Aristada) [108,109]. One of the most significant clinical advances with aripiprazole is the approval of a unique delivery system, where a single oral dose of 30 mg aripiprazole can be given with 1064 mg of the LAI, aripiprazole lauroxil (Aristada^®^), along with an immediate release nanomolar preparation (Aristada Initio^®^) to achieve a clinically effective range of plasma levels of aripiprazole in a single day [110]. Administration of this aripiprazole combination ensures adequate aripiprazole levels in hospitalized patients at hospital discharge to prevent psychotic relapse during the most vulnerable post-hospital discharge. The real-world implication of LAIs is highlighted by two long-term outcome studies involving aripiprazole and paliperidone LAIs that reported reduced hospitalization rates [111,112]. Although some studies comparing oral APMs and LAIs have yielded some inconsistent results, the pooled data from a meta-analysis based on 42 studies showed the clinical superiority of the two LAIs over oral APMs in reducing hospitalization rates [113]. Based on findings from a large prospective study from Sweden over seven years, LAIs had the highest rates of relapse prevention and lowest rates of rehospitalizations in patients with schizophrenia [114]. Of note, only one oral APM, clozapine, could match these results from LAIs.

Cariprazine is indicated for treating schizophrenia and acute manic or mixed episodes associated with bipolar I disorder as a monotherapy [115,116]. It is the only antipsychotic medication that received approval for schizophrenia and bipolar I disorder at the same time [117]. Cariprazine is a novel antipsychotic with unique pharmacodynamic and pharmacokinetic properties [118,119]. Regarding pharmacodynamics, cariprazine is similar to other SGAMs exhibiting antagonistic activity at 5HT2A receptors [97]. In addition, it is similar to aripiprazole and brexpiprazole in demonstrating partial agonist activity at D2 and D3, and 5HT1A receptors (Table 1). However, cariprazine is the only partial agonist at D2 receptors that exhibits a higher affinity for the D3 than the D2 receptor. While this unique property continues to be highlighted throughout the literature, the clinical significance of this remains unknown. Cariprazine also has moderate histamine antagonism, low alpha-1 antagonism, and no significant affinity for muscarinic cholinergic receptors (Table 1) [119]. Cariprazine has two biologically active metabolites, desmethyl and didesmethyl cariprazine [119], of which the latter is primarily accountable for later and long-term efficacy and tolerability [117]. The most common treatment-emergent adverse events in the cariprazine group included insomnia, EPS, akathisia, sedation, nausea, dizziness, and constipation [118]. Cariprazine treatment was comparable to the placebo in weight gain and metabolic parameters (Figure 3), vital signs, sedation, prolactin, or QTc interval [118]. In addition, cariprazine-induced akathisia was reported as mild to moderate (Figure 1), which suggests that cariprazine may be slightly better than aripiprazole in metabolic adverse effects and akathisia [118]. However, akathisia was dose-related, and higher cariprazine doses than 3 mg/day resulted in a higher risk for akathisia. The only treatment-related serious adverse event was supraventricular tachycardia in one study subject, which resolved after the drug discontinuation [118].

Brexpiprazole is the latest D2 receptor partial agonist that received approval for the treatment of schizophrenia and for adjunctive use for MDD in patients who demonstrated inadequate response to standard antidepressant therapy [120,121]. The recommended dose for the treatment of schizophrenia is 2–4 mg/day [122]. Like aripiprazole, it is a partial agonist at D2 and 5HT1A receptors and an antagonist at serotonin 5HT2A receptors (Table 1) [123]. However, brexpiprazole displays less intrinsic activity at D2 receptors and, coupled with actions at 5HT1A, 5HT2A, and noradrenergic α1 receptors that are at least as potent as its action at D2 receptors, it is predicted to demonstrate a lower propensity to activating adverse events, such as akathisia and other EPS than aripiprazole (Figure 1) [123]. Clinically, brexpiprazole produced more statistically significant improvements in schizophrenia symptomatology and psychosocial functioning than placebo in adults with acute schizophrenia [124]. In long-term trials, brexpiprazole reduced the time to relapse compared with placebo, with continued improvement in symptoms and functioning [125]. Brexpiprazole was generally well tolerated with a relatively low incidence of activating and sedating adverse effects, small changes in QT interval and metabolic parameters that were not clinically significant, and moderate weight gain (Figure 3 and Figure 4) [124].

## 5. Multimodal Antipsychotic Medication

Much more recently, another novel APM, lumateperone, has been FDA-approved, which appears to behave entirely differently from earlier APMs [126]. Due to its novel and unique mechanism of action beyond dopamine serotonin antagonism (i.e., SGAMs) and partial agonism at D2 receptors, it can be labeled as a multimodal APM [127]. Most importantly, lumateperone is the only APM with a differential effect on pre- and post-synaptic D2 receptors [127]. It is a partial agonist at the presynaptic D2 receptors, which are autoreceptors, and an antagonist at the post-synaptic D2 receptors. Since partial agonism at the presynaptic receptors decreases dopamine release from the presynaptic neurons, even 40% of post-synaptic D2 receptor blockade with lumateperone is effective for antipsychotic effects as opposed to 60–80% needed with other APMs [128,129]. In addition, lumateperone has also been found to modulate glutamatergic effects to further contribute to its antipsychotic efficacy [127]. The clinical highlight of lower D2 receptor blockade is the much lower risk for D2 receptor-mediated adverse effects, such as EPS and hyperprolactinemia (Figure 1) [130,131]. These mechanisms of action may also explain lumateperone efficacy in affective disorders as well [132]. Although aripiprazole, cariprazine, and brexpiprazole are also partial agonists at the D2 receptors, they don’t differentiate between pre-and post-synaptic D2 receptors and thus require higher D2 receptor occupancy to mediate antipsychotic effects (Table 1). Although lumateperone may not have any significant edge in efficacy over prior APMs, it may be one of the most tolerable APMs currently available.

## 6. Future Directions

It is evident from the prior discussion in this review that there are countless unmet needs in treating schizophrenia. One of the foremost needs is to conduct long-term and significantly large sample randomized controlled trials (RCTs) to determine the duration of treatment and the maintenance of antipsychotic doses. Although it is going to be challenging to conduct such expensive and multicenter trials, multicenter collaborations, partnerships with industry and federal government, and academic consortia would be how these goals can be accomplished. The large sample sizes provide the power to conduct subgroup analyses to control for heterogeneity in schizophrenia and treatment resistance and to generate new hypotheses for future investigations [133]. In addition, there is a need for a paradigm shift from the dopaminergic to glutamatergic and GABAergic hypotheses, the brain’s two most ubiquitous neurotransmitter systems, to address negative and cognitive symptom domains not managed by the current APMs. Furthermore, there needs to be a multifaceted translational neuroscience approach utilizing neuroimaging, pharmaco genomics, proteomics, metabolomics, and epigenomics to develop novel drug targets, predict drug response, and reduce adverse effects, promoting personalized medicine [134]. One of the most significant advantages of developing more efficacious and tolerable personalized antipsychotic treatments is improved medication adherence, which can be optimized by promoting long-acting injectables.

## Figures and Tables

**Figure 1 biomedicines-11-00130-f001:**
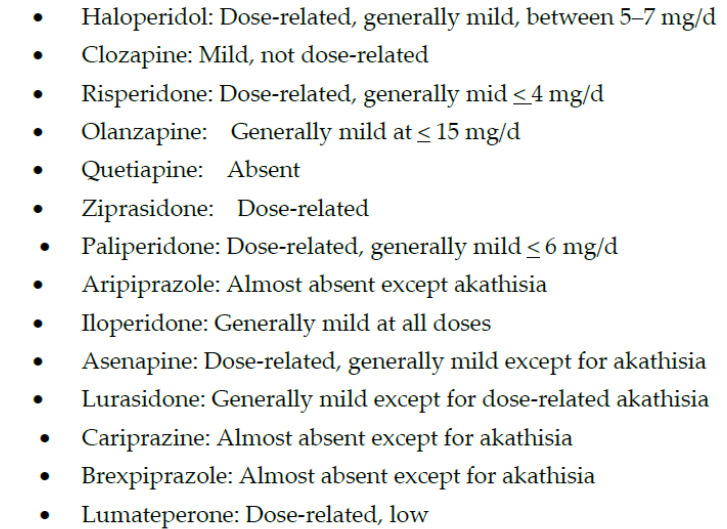
Extrapyramidal Symptoms (EPS) with antipsychotic medications.

**Figure 2 biomedicines-11-00130-f002:**
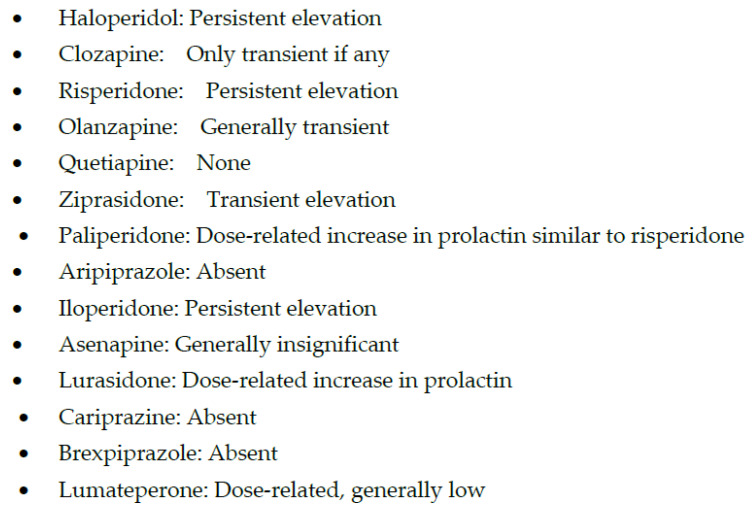
Hyperprolactinemia with antipsychotic medications.

**Figure 3 biomedicines-11-00130-f003:**
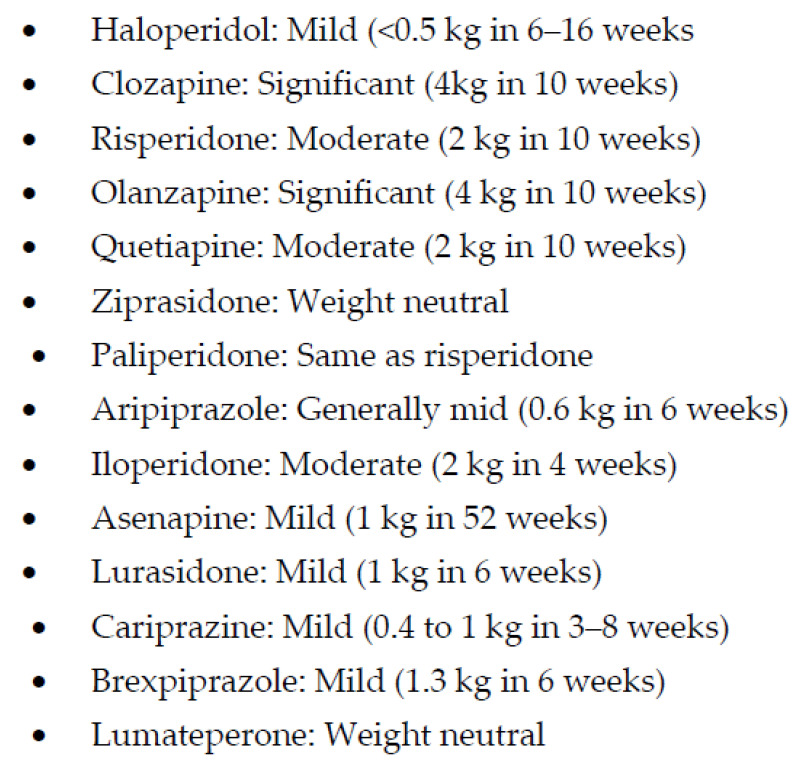
Weight gain with antipsychotic medications.

**Figure 4 biomedicines-11-00130-f004:**
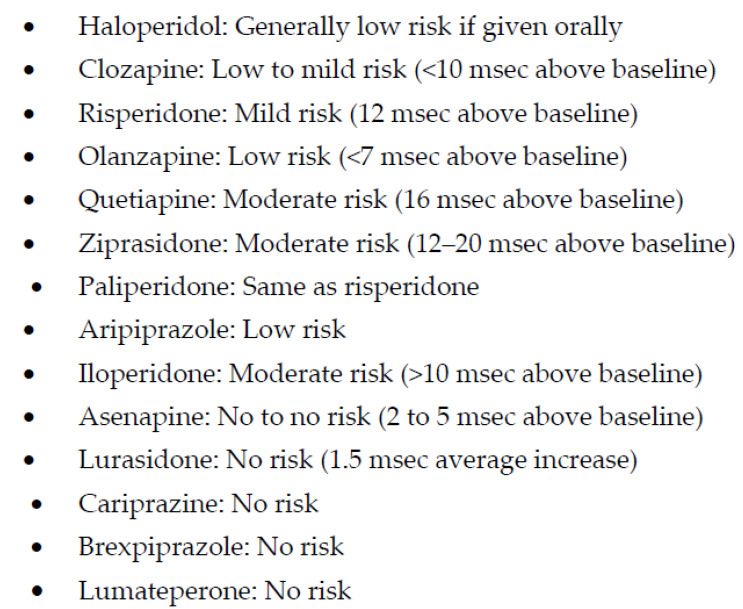
QTc prolongation with antipsychotic medications.

**Table 1 biomedicines-11-00130-t001:** Molecular targets and binding affinities (Ki, nM) for antipsychotic medications.

Receptors	HAL	CLZ	RIS	OLZ	QUE	ZIP	PALI	ARIP	ILO	ASEN	LUR	CAR	BREX	LUMA
D2	0.7	126	4	11	160	5	4.6	0.45	6.3	1.3	1.68	0.49	0.3	32
5HT1A	>1000	875	210	>10,000	>1000	3	617	4.4	168	2.5	6.75	2.6	0.12	-
5HT2A	45	16	0.5	4	295	0.4	1.1	3.4	5.6	0.06	2.03	18.8	0.47	0.54
5HT2C	>10,000	16	25	23	>1000	1	48	15	-	0.03	415	134	12–34	-
D3	-	-	-	-	-	-	3.5	-	7.1	-	15.7	0.08	1.1	--
5HT7	-	18	6.6	110	310	6	2.7	10	22	0.13	0.5	111	3.7	-
α1	6	7	0.7	19	7	10	2.5	57	0.36	1.2	47.9	155	0.17	73
α2	-	16	11	210	350	400	3.9	38	-	1.2	10.8	-	0.59	-
H1	440	6	20	7	11	47	19	61	437	1.0	>1000	23.2	19	>1000
M1	>1000	1.9	>10,000	1.9	120	>1000	>10,000	>10,000	>1000	>1000	>1000	>1000	>1000	>1000

Values are K_i_ (nM). The smaller the value, the more strongly the drug binds to the site. HAL = Haloperidol; CLZ = Clozapine; RIS = Rsiperidone; OLZ = Olanzapine; QUE = Quetiapine; ZIP = Ziprasidone; PALI = Palipreidone; ARIP= aripiprazole; ASEN = Asenapine; LUR = Lurasidone; CAR = cariprazine; BREX = Brexpiprazole; LUMA = Lumateperone. Arnt & Skarsfeld, 1998 [22]; Bymaster et al. 1996 [23]; Seeger et al. 1995 [24]; Schotte et al. 1996 [25], Maeda et al. 2014 [26]; Roth & Driscol, 2021 https://pdsp.unc.edu/databases/pdsp.php, accessed on 6 December 2022; Herman et al. 2018 [27]; Citrome 2010 [28] ; Citrome 2013 [29]; Corena-McLeod 2015 [30].

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
