# Peer review of "Seventy Years of Antipsychotic Development: A Critical Review"

_biomedicines, 2023, doi:10.3390/biomedicines11010130_

Round 1

Reviewer 1 Report

1. I suggest to provide a short intro with a table or a figure demonstrating receptor targets for known antipsychotics of the first and second generation. This would remind the Reader differences between both groups. Also, since the paper is lack of any figures and /or tables, this would "break" the manuscript text.

2. Some sentences should be rewritten as they are too long and its hard to follow.

3. The Author written that changes in CYP450 isoformes (such as CYP2D6 or CYP3A4) may resulted in pharmacodynamic changes of clozapine. However, to my knowledge, these two types of CYP450 enzymes are responsible for the liver metabolism of almost 70% of all clinically available drugs. Therefore, this action is not limited to clozapine only. Moreover, also pharmacokinetic interactions may occur.

4. The Author described ziprasidone. Since thus drug structurally differs from other, as this is a chimeric drug (hybrid), the Author should characterize it more detailed. A nice paper was presented by Kleczkowska who showed potent hybrid compounds in psychiatric disorders. 

5. The Author introduced a section "partial agonists" of dopaminergic D2 receptors. However, there is no such a division while demonstrating first and second generation. Therefore, I'm not sure why did the Author focus on partial agonists and specifically at DRD2 ? In line with this, why  there is no info about inverse agonists? 

Author Response

December 19, 2022

To

The Editors,

Biomedicines

Subject: Revision of "Seventy Years of Antipsychotic Development: A Critical Review."

Reviewer 1:

Critique 1:  I suggest providing a short intro with a table or a figure demonstrating receptor targets for known antipsychotics of the first and second generation. This would remind the Reader differences between both groups. Also, since the paper lacks figures and/or tables, this would "break" the manuscript text.

Response 1: I have added the Table 1 with Ki vlaues for newer antipsychotic medications with haloperidol from older antipsychotic drugs. In addition, I have also added 4 Figures showing list of common adverse effects and differential efffects of antipsychotic medications. 

Critique 2: Some sentences should be rewritten as they are too long and hard to follow.

Response: Have revised and rephrased manuscript to break down long sentences.

Critique 3. The author wrote that changes in CYP450 isoforms (such as CYP2D6 or CYP3A4) might result in pharmacodynamic changes in clozapine. However, to my knowledge, these two types of CYP450 enzymes are responsible for the liver metabolism of almost 70% of all clinically available drugs. Therefore, this action is not limited to clozapine only. Moreover, also pharmacokinetic interactions may occur.

Response 3:

Adding detailed metabolic pathways for all antipsyhotic drugs will consume lot of space. But clinically relevant metabolic information for all antipsychotic medications has been added.

Critique 4. The author described ziprasidone. Since this drug structurally differs from others, as this is a chimeric drug (hybrid), the author should characterize it in more detail. An excellent paper was presented by Kleczkowska, who showed potent hybrid compounds in psychiatric disorders. 

Response 4: We have added a paragrpah on this important tpoic of chimeric nature of ziprasidone. Please refer to lines 269 to 276 on pages 13.

Critique: 5. The Author introduced a section called "partial agonists" of dopaminergic D2 receptors. However, there is no such division while demonstrating first and second generation. Therefore, I'm not sure why the author focused on partial agonists, specifically at DRD2. In line with this, why is there no info about inverse agonists? 

Response 5: I have added more clarification on the reasons of not inclduing partial agonsists and lumaterpeone under the second generation antipsychotic drugs. I have also briefly discussed partial agonism at 5HT1A receptors with second-generation antipsychotic medications (where applicable) as well as all partial agonsists.

We appreciate the positive feedback from the reviewer for the valuable comments that have significantly improved the manuscript in its clarity and take-home message.

Thank you

Sincerely,

Mujeeb U Shad, M.D., M.S.C.S.

Adjunct Professor of Psychiatry,

University of Nevada, Las Vegas,

Adjunct Professor of Psychiatry,

Touro University Nevada College of Osteopathic Medicine, Las Vegas, NV.

Psychiatry Residency Program Director,

Valley Health System, Las Vegas, NV.

Reviewer 2 Report

This is a comprehensive and well-written review on antipsychotic medications their place in the history of psychopharmacology.

However, I feel that some more information concerning currently available evidence on their role in clinical practice should be reported, and I have a few conceptual suggestions for the Author in this regard.

1.     Some more clinical, real-world evidence should be considered (https://doi.org/10.1001/jamapsychiatry.2017.1322).

2.     When dealing with antipsychotic development, their long-acting formulation cannot be forgotten in view of their wide application and great usefulness in clinical practice. However, the Author only provides little information in this regard. He should please refer to:

a.     https://doi.org/10.1016/j.psychres.2022.114405 for real-world comparisons of LAI use in a clinical sample;

b.     https://doi.org/10.1093/schbul/sbx090 for comparison between oral and LAI antipsychotics use.

3.     Except for the approval note for cariprazine, there is almost no mention of bipolar disorder, which is actually one on the most important and frequent occurrence of antipsychotic treatment. Please refer to:

a.     https://doi.org/10.1038/s41380-021-01334-4 for antipsychotic use for the treatment of acute mood episodes.

b.     https://doi.org/10.1038/s41380-020-00946-6 for oral and https://doi.org/10.1007/s00406-022-01522-5 for long-acting antipsychotics in the maintenance treatment of bipolar disorder.

Author Response

December 19, 2022

To

The Editors,

Biomedicines

Subject: Revision of "Seventy Years of Antipsychotic Development: A Critical Review."

Reviewer 2

Critique 1: However, I feel that some more information concerning currently available evidence on their role in clinical practice should be reported, and I have a few conceptual suggestions for the author in this regard. Some more clinical, real-world evidence should be considered (https://doi.org/10.1001/jamapsychiatry.2017.1322).

Response 1: I have added two crucial landmark studies with real-world implications for LAIs. Please refer to lines 377 to 384 on page 18.

Critique 2:     When dealing with antipsychotic development, their long-acting formulation cannot be forgotten in view of their wide application and great usefulness in clinical practice. However, the author only provides little information in this regard. He should please refer to the following:

  1. https://doi.org/10.1016/j.psychres.2022.114405for real-world comparisons of LAI use in a clinical sample;
  2. https://doi.org/10.1093/schbul/sbx090for comparison between oral and LAI antipsychotics use.

Response 2: As stated above, both these references have been added to the revised manuscript. Please refer to lines 377 to 384 on page 18.

Critique 3: Except for the approval note for cariprazine, there is almost no mention of bipolar disorder, which is actually one of the most important and frequent occurrences of antipsychotic treatment.

Response 2:

The paper is already quite lengthy, and adding bipolar studies will make it even longer. However, I have redefined the goal of this paper to only address schizophrenia treatment. Please refer to lines 121-122 on page 6.

We appreciate the positive feedback from the reviewers and their valuable comments that have significantly improved the manuscript in its clarity and take-home message.

Thank you

Sincerely,

Mujeeb U Shad, M.D., M.S.C.S.

Adjunct Professor of Psychiatry,

University of Nevada, Las Vegas,

Adjunct Professor of Psychiatry,

Touro University Nevada College of Osteopathic Medicine, Las Vegas, NV.

Psychiatry Residency Program Director,

Valley Health System, Las Vegas, NV.

Round 2

Reviewer 1 Report

The paper was now improved greatly, therefore in my opinion it is suitable for publication.

Author Response

Thank you 

Reviewer 2 Report

No further comments.

Author Response

No response needed